# Evidence-based practice utilization and associated factors among nurses working in Amhara Region Referral Hospitals, Ethiopia

Zewdu Bishaw Aynalem[1]*, Kassahun Gebeyehu Yazew[2], Mignote Hailu Gebrie[3]

1 Department of Nursing, Bahir Dar Health Science College, Bahir Dar, Ethiopia, 2 Department of Medical Nursing, School of Nursing, College of Medicine and Health Sciences, University of Gondar, Gondar, Ethiopia, 3 Department of Surgical Nursing, School of Nursing, College of Medicine and Health Sciences, University of Gondar, Gondar, Ethiopia

* zest7@yahoo.com

## Abstract

### Background

Applying evidence-based practice during care provision is essential because it improves the quality of care, reduces health care costs, and increases patient and family satisfaction. However, information on evidence-based nursing practice and associated factors were not well studied and documented in the study area. Hence, this study aimed to assess utilization and associated factors of evidence-based practice among nurses working in Amhara Region Referral Hospitals, Ethiopia.

### Methods

An institution-based cross-sectional study was conducted from March 18 to April 16, 2019, in Amhara Region Referral Hospitals. A simple random sampling technique was used to select 684 respondents. Data were collected using a pretested and self-administered questionnaire. Data were entered into Epi Info version 7.1.2.0 and exported to SPSS version 22.0 for analysis. The bivariable analysis was used primarily and variables with p-value < 0.2 were further examined using a multivariable logistic regression model to control confounders. Then, variables' p-value < 0.05 with 95% CI was used to determine associated factors.

### Results

From 684 proposed nurses, 671 of them completed the questionnaire giving 98.1% response rate. Of these, 55% (95% CI: 51.2, 58.9) of them had good evidence-based practice utilization. Variables including single (AOR = 1.662: 95% CI: 1.089–2.536), fewer work experience (AOR = 1.849: 95% CI: 1.049–3.257), good knowledge (AOR = 2.044: 95% CI: 1.406–2.972), effective communication skill (AOR = 2.537: 95% CI: 1.744–3.689), EBP training (AOR = 3.224 95% CI: 1.957–5.311), internet access (AOR = 1.655: 95% CI: 1.119–2.448) and evidence-based guideline availability (AOR = 1.827: 95% CI: 1.249–2.673) were found to be predictors of evidence-based practice utilization.

**Data Availability Statement:** All relevant data are within the paper and its Supporting Information files.

**Funding:** Amhara Regional Health Bureau paid per diem for data collectors during data collection. The funder had no role in study design, data collection and analysis, decision to publish, or preparation of the manuscript.

**Competing interests:** The authors have declared that no competing interests exist.

**Abbreviations:** AOR, Adjusted Odds Ratio; ARRH, Amhara Region Referral Hospitals; CI, Confidence Interval; COR, Crude Odds Ratio; EBP, Evidence-Based Practice; ETB, Ethiopian Birr; IQR, Inter Quartile Range.

## Conclusions

The study revealed that evidence-based practice utilization among nurses is low. Availing evidence-based guidelines in the work area, improving facilities' internet access, and building nurses' evidence-based practice competencies through either by giving separate training or incorporating as part of the curriculum would improve its utilization.

## Introduction

Evidence-Based Practice (EBP) is an approach to a clinical practice whereby clinicians integrate current best evidence with their clinical expertise to make decisions for a specific client by considering his or her preferences and values [1, 2]. As such, it is a decision-making process for practice through the conscientious, explicit, and judicious use of the best available evidence from multiple sources [3, 4].

By evidence, we mean clinically relevant research findings including empirical evidence from randomized controlled trials; evidence from other scientific methods such as descriptive and qualitative research; as well as the use of information from case reports, scientific principles, and expert opinion [5]. By clinical expertise, we mean the knowledge and skill of nurses about their field of study. Nurses have knowledge and skill that are developed from their academic education and clinical experiences. Undoubtedly, nurses have skills in how to communicate with patients and how to elicit their feelings. Thus, it's about integrating their knowledge and skill during care provision [6].

Patient preference and value refer to patient perspectives, beliefs, expectations, and goals for health and life that influence choices; these must all be weighed against the evidence. Patients may opt for one treatment over another because of comfort, cost, convenience, or other beliefs. Therefore, it is about listening to them and using these as input during their health care decision-making [7].

Though EBP is used across various professions as an approach to professional practice, it is rapidly growing in the fields of nursing [8]. Evidence-based practice positively influences the practice of nurses and enables them to shift from tradition to scientific-based practice [9]. Similarly, it results in reduced costs, improved patient outcomes, and serves as a standard for quality patient care [10, 11], increases patient and family satisfaction, and contributes to professional development [8]. Again, it results in high job satisfaction [12], increases working efficiency, and reduces overtime [13], and fills the gap between theories and practice [14]. Honestly, it improves the health care workers' hand hygiene practice in low-resource countries like Uganda and Ethiopia [15, 16].

Despite having various positive effects on patient outcomes, nurses do not regularly utilize EBP [10]. The reason for this is that nurses face numerous challenges in converting evidence into practice [9, 17, 18]. Among these; lack of source, lack of time, inadequate skill, lack of training, and lack of knowledge took the first [19]. Similarly, evidence indicated nurses who use EBP have been shown to make better decisions in service delivery [20]. However, there are many factors to implementing EBP, both at the individual and organizational level [8]. Barriers at the level of the individual include the nurse's ability to work with computers [21], attitude [22], and knowledge about EBP [23]. Additional organizational barriers include lack of time to read literature [24], lack of internet access [25], and absence of a nurse manager who leads and facilitates EBP utilization at the work area [9, 26].

By the same token, although, EBP is considered to be the gold standard in nursing practice, studies from different countries have reported that nurses EBP utilization is low: in Norway

nurses' practice towards EBP was to a small extent and most nurses used their experiences instead of scientific evidence during care provision; in Australia, the use of research focused EBP is one-third; in Iran only 41% and in Nepal, 49% of nurses used EBP during their health care delivery [22, 24, 27–29].

Similarly, studies in Africa have shown that a significant amount of nurses had poor EBP utilization. About 54% in Zambia [30], 25.3% in Ghana [31], 30.9% in Nigeria [32], and 12% of nurses in Kenya [33] had utilized research findings during their clinical practice. Evidence in this region showed that even if EBP is a new health care approach that is effective and could reduce the burden of tuberculosis (TB), HIV/AIDS, malaria, and other infectious diseases by guiding preventive and curative aspects, it has not yet widely implemented [34, 35]. It was against this background that the 2012 Kigali declaration was signed to improve the use of evidence for clinical decision-making [36].

Ethiopia, similar to other African countries, does not have a well-described report on evidence-based nursing practice. Few studies are available that focus on this issue [23, 37–39]. To bring evidence-based practice onto the agenda at national, regional, and local levels, Ethiopia arranged a workshop on evidence-based healthcare for different health professionals in Addis Ababa [40]. Despite the effort underway, studies highlighted that nursing care in Ethiopia is not evidence-based rather it relies on experience, tradition, common sense, and untested theories [37, 39]. Also, at all levels of the health systems, there is a little culture of using research evidence during decision making [41].

To this end, previous studies that were done in the study area [25, 42] focus on EBP utilization among different health care professionals which couldn't specifically show nurses' gap. Both of them were small-scale surveys limited to particular corners of the region. Hence, they did not provide a full picture of the extent of evidence-based nursing practice on a regional level. Moreover, these studies didn't give more emphasis on the association between communication skills and nurses' EBP utilization. Therefore, this study was conducted to determine EBP utilization and identify factors that affect it among nurses working in Amhara Region Referral Hospitals (ARRH), Ethiopia.

## Materials and methods

### Study design, setting, and participants

An institution-based cross-sectional study was conducted from March 18 to April 16, 2019, among nurses working in ARRH. The Region is located in the Northwestern and Northcentral parts of Ethiopia. It has eleven administrative zones. Currently, the region has sixty-seven public hospitals, five of which; University of Gondar, Felege Hiwot, Dessie, Debre Birhan, and Debre Markos are Referral Hospitals. Each Referral Hospital serves 3.5–5 million people [43]. During the study period, a total of 458, 390, 254, 140, and 223 nurses were working at the above-mentioned Referral Hospitals respectively. Of these, 189, 184, 128, 74, 68, and 41 were assigned in the surgical, medical, ambulatory, pediatrics, emergency, and critical care units respectively. Nurses who had work experience of 06 months and above were included in the study. However, nurses who were on maternity or annual leave or those who were seriously ill during the data collection period were excluded.

## Objectives

The objectives of this study were to determine evidence-based practice utilization and to identify factors associated with EBP utilization among nurses working in Amhara Region Referral Hospitals, Ethiopia,

**Table 1. Sample size calculation for factors associated with utilization of EBP from the previous study among nurses working in ARRH, Ethiopia, 2019.**

| Serial No | Predictor variables | Non-exposed (p1) | Exposed (p2) | Confidence level (%) | Power (%) | Odds ratio | Sample size | 10% non-response rate | Final sample size |
|---|---|---|---|---|---|---|---|---|---|
| | Knowledge about EBP | 42.9 | 58.7 | 95 | 80 | 1.89 | 338 | 34 | **372** |
| | Internet access | 48.9 | 60.4 | 95 | 80 | 1.59 | 622 | 62 | **684** |
| | Self-reported availability of time | 43.9 | 59.9 | 95 | 80 | 1.9 | 330 | 33 | **363** |

## Protocols

http://dx.doi.org/10.17504/protocols.io. [https://orcid.org/0000-0003-2419-1502]

## Sample size

The sample size was calculated for both objectives. For objective one, it was calculated using the single population proportion formula by considering the following assumptions: 95% confidence interval, 51.8% proportion [37], and 5% marginal error. By adding a 10% nonresponse rate, the final sample size was 422.

For the second objective, it was also calculated by taking three predictor variables from a previous study [25] using Epi Info version 7.1.2.0 software (**Table 1**).

The largest sample size from the table was 684, which was higher than the sample size calculated from the single population proportion formula above. Thus, the minimum adequate sample size for this study was 684.

## Sampling procedures and techniques

To select 684 nurses from the total five referral hospitals in Amhara Region, all hospitals were first listed down with their respective nurse numbers after which sample size was proportionally allocated to each hospital. Then, the sampling frame was prepared for each hospital by having lists of nurses from the hospitals' human resource management. Finally, eligible nurses of each hospital were selected by a simple random sampling technique.

## Instrument and measurement

Data were collected by using a pre-tested, structured, and self-administered questionnaire, which was adapted from different works of literature [44–48]. The questionnaires were prepared in English and organized into four main sections: socio-demographic, individual-related factors, organization-related factors, and evidence-based practice questions. Ten BSc nurses and five MSc nurses were recruited as data collectors and supervisors respectively. They got a half-day training on how to facilitate the data collection process, about the aim of the study, confidentiality, and verbal consent.

A pre-test was conducted in a similar setting two weeks before the actual data collection time on 10% of the sample size. Supervisors closely followed the data collection process. Continuous supervision was also made by the principal investigators throughout the data collection period. Every day after data collection, questionnaires were reviewed and checked for completeness. Then, necessary feedback was offered to data collectors in the next morning.

Previous studies proved that the EBP utilization sub-scale is valid and reliable. In two studies, its Cronbach's alpha was documented as 0.93 [44] and 0.847 [45]. But, for the sake of contextualization, the reliability of the tool was checked via Cronbach's alpha value using the data obtained from the pretest and found to be 0.91, 0.92, 0.85, and 0.92 for knowledge, attitude,

utilization, and communication sub-scales respectively. Similarly, to ensure its validity, the developed tool was given to two experts in the field for critiquing.

The outcome variable of this study was EBP utilization (good/poor) among nurses. The independent variables were sociodemographic characteristics, individual factors, and organizational factors. To measure nurses' EBP utilization, six questions were asked. Each question had a 5-point Likert scale (1 = never, 2 = sometimes, 3 = usually, 4 = often, and 5 = always) with a minimum score of 6 and a possible maximum score of 30. According to the Shapiro Wilk test, utilization score was not approximately normally distributed (W statistic = 0.977, df = 671 and p-value < 0.001). Hence, the median with interquartile range (IQR) of utilization score, 16 (±9), was used as a cutoff point. Nurses who scored equal and above the cutoff point were categorized as having "good EBP utilization" otherwise "poor EBP utilization".

A total of 17 questions were asked to measure nurses' knowledge about EBP, with each of them having a 3-level response (1 = no, 2 = somewhat, 3 = yes). 'No' response was coded as '1', 'somewhat' responses as '2', and 'yes' responses as '3'. Each respondent's total EBP knowledge scores with a possible minimum score of 17 to a maximum of 51 were calculated. Using the Shapiro–Wilk test, the nurses' knowledge score was checked for normal distribution. Therefore, knowledge about EBP was not normally distributed (W statistic = 0.976, df = 671 and p-value <0.001). The median (IQR) of the knowledge score about EBP was 34 (±13). Nurses having a knowledge score equal and above the median score were considered as having "good knowledge" and those below as having "poor knowledge".

Eleven questions were asked to measure nurses' attitudes towards EBP. Each attitude question had a five-point rating scale (1 = strongly disagree, 2 = disagree, 3 = neutral, 4 = agree and 5 = strongly agree). Each respondent had a possible total attitude score ranging from 11 to 55. The normal distribution of the respondent's attitude score was checked by the Shapiro Wilk test and found not normally distributed (W statistic = 0.888, df = 671, and p-value <0.001). The median (IQR) attitude score towards EBP was 43 (±9). Nurses who scored equal and above the median were labeled as having a "favorable attitude" if not as "unfavorable attitude".

Sixteen questions were adapted from a previous study to measure the effectiveness of nurses' communication skills with the patient. Each item had a 5-point Likert scale (1 = strongly disagree, 2 = disagree, 3 = neutral, 4 = agree, and 5 = strongly agree) with a minimum score of 16 and a maximum score of 80. After checking for normal distribution using the Shapiro–Wilk test, the median (IQR) communication score, 59 (±25), was set as a cutoff point because the communication score was not normally distributed (W statistic = 0.957, df = 671 and p-value < 0.001). Respondents who scored equal and above the median score were classified as having "effective communication" otherwise "ineffective communication".

## Statistical analysis

Data were cleaned, coded, and entered into Epi-Info version 7.2.1.0 software and then exported to SPSS version 22.0 for analysis. The result was interpreted and presented using appropriate tables, graphs, and charts. Descriptive parameters, such as mean, mode, and median for continuous variables and frequencies and percentages for categorical variables were calculated. The Binary logistic regression model was fitted to identify factors associated with EBP utilization. Variables with a p-value < 0.2 in the bi-variable logistic regression model were entered into a multivariable logistic regression model to control confounders. Model fitness was checked using a Hosmer-Lemeshow goodness-of-fitness test (p-value = 0.360). Finally, the statistical significance was declared at the p-value less than 0.05 with a 95% Confidence interval (CI).

### Ethical approval

The study was approved by the School of Nursing ethical review committee on behalf of the Institutional review board of University of Gondar by an approval number of S/N/1600/06/2011. Before beginning data collection, a permission letter was obtained from the School of Nursing and was submitted for each Referral Hospital. From each participant, verbal informed consent was obtained after clearly describing the purpose of the study. Again, nurses were informed of their full right to skip or ignore any questions or not to participate in the study.

## Results

### Socio-demographic characteristics of the respondents

Out of 684 proposed nurses, 671 of them completed the questionnaire, which makes a response rate of 98.1%. Almost half of the nurses 342 (51%) were male. The largest proportion, 442 (65.9%), of them were under the age category of 21–30 years and their median (IQR) age was 29 (±7) years. The majority of the respondents were married 447 (66.6%) and Bachelor's degree holders 571 (85.1%). Concerning education completion: the vast majority, 598 (89.1%), of them completed their education from government institutions whereas the rest were from some private university colleges. The majority of respondents were from surgical and medical units each accounted for 27.1% and the least, 6.1%, of them were from the critical care unit. Based on quartile classification, the median (IQR) monthly salary of the nurses was 5294 (±2461) Ethiopian Birr (**Table 2**).

### Individual related factors

Of the total 671 nurse respondents, 517 (77%) of them didn't take EBP training either as part of their curriculum or separately, 42.3% of them had no research experience and 49.5% of them had difficulty in interpreting research findings. As well, about 46.1%, 48.9%, and 49.6% of them had poor EBP knowledge, unfavorable attitude towards EBP, and ineffective communication skill respectively. Shockingly, 145 (21.6%) of them reported that they lack the willingness to utilize EBP, and 33.2% of them had no basic computer skills in how to search for evidence.

### Organizational related factors

Four hundred nine (61%) of nurses reported that evidence-based guidelines are available at the workplace. But, nearly half (47.2%) of them reported that there is no internet access to utilize EBP. Similarly, 409 (61%) and 367 (54.7%) nurses reported that there is no enough time and no nurse manager who leads and facilitates EBP utilization at the work area respectively. Furthermore, fifty-seven percent of respondents reported that they have a patient load (>6:1 patient to nurse ratio), 54.5% and 54.2% of them reported that there is no library in the hospital and no computer in their work area/department respectively.

### Evidence-based practice utilization

In this study, the magnitude of nurses who had good evidence-based practice utilization was found to be 55% (95% CI: 51.2, 58.9). The median (IQR) of nurses' utilization score was 16 (±9). About 131 (19.5%) nurses frequently formulate clinical questions. Additionally, 177 (26.4%) of them usually integrate evidence that they got with patient values and their skills. But, only 7.3% of them appraise the literature always and 14.3% of them never evaluate the outcome of their practice (**Table 3**).

**Table 2. Sociodemographic characteristics of nurses working in ARRH, Ethiopia, 2019 (n = 671).**

| Variables | Category | Numbers | Percent |
|---|---|---|---|
| Sex | Female | 329 | 49 |
| | Male | 342 | 51 |
| Age (in years) | 21–30 | 442 | 65.9 |
| | 31–40 | 159 | 23.7 |
| | 41–50 | 58 | 8.6 |
| | 51–60 | 12 | 1.8 |
| Marital status | Single | 224 | 33.4 |
| | Ever-married | 447 | 66.6 |
| Educational status | Diploma Nurse | 62 | 9.2 |
| | BSc Nurse | 571 | 85.1 |
| | MSc Nurse | 38 | 5.7 |
| Place of education completion | Government | 598 | 89.1 |
| | Private | 73 | 10.9 |
| Work experience (in years) | <5 | 255 | 38.0 |
| | 5–10 | 140 | 20.9 |
| | >10 | 116 | 17.3 |
| Working unit | Critical care unit | 41 | 6.1 |
| | Pediatrics unit | 74 | 11 |
| | Medical unit | 182 | 27.1 |
| | Surgical unit | 182 | 27.1 |
| | Emergency unit | 64 | 9.5 |
| | Ambulatory unit | 128 | 19.1 |
| Nurses monthly salary (ETB) | <4650 | 171 | 25.5 |
| | 4650–5294 | 175 | 26.1 |
| | 5294–7111 | 199 | 29.7 |
| | >7111 | 126 | 18.8 |

## Factors affecting evidence-based practice utilization of nurses

A multivariable binary logistic regression analysis was performed to identify factors associated with EBP utilization among nurses. Accordingly, seven variables were found to be statistically associated with EBP utilization after adjusting for confounders. These were marital status, work experience, knowledge about EBP, communication skills, training about EBP, evidence-based guideline availability, and internet access.

Nurses who were single in marital status were 66.2% higher (AOR = 1.662: 95% CI: 1.089, 2.536) to have good EBP utilization compared to ever-married nurses. Nurses who had worked

**Table 3. Frequency of EBP utilization among nurses working in ARRH, Ethiopia, 2019.**

| Activities | Never | | Sometimes | | Usually | | Often | | Always | |
|---|---|---|---|---|---|---|---|---|---|---|
| | N | % | N | % | N | % | N | % | N | % |
| Formulating clinical question | 84 | 12.5 | 211 | 31.4 | 175 | 26.1 | 131 | 19.5 | 70 | 10.4 |
| Searching evidence | 106 | 15.8 | 219 | 32.6 | 170 | 25.3 | 112 | 16.7 | 64 | 9.5 |
| Appraising evidence | 149 | 22.2 | 210 | 31.3 | 155 | 23.1 | 108 | 16.1 | 49 | 7.3 |
| Integrating with expertise | 104 | 15.5 | 191 | 28.5 | 177 | 26.4 | 119 | 17.7 | 80 | 11.9 |
| Evaluating outcomes | 96 | 14.3 | 188 | 28.0 | 152 | 22.7 | 129 | 19.2 | 106 | 15.8 |
| Sharing outcomes | 109 | 16.2 | 213 | 31.7 | 131 | 19.5 | 136 | 20.3 | 82 | 12.2 |

**Table 4. Binary logistic regression analyses of socio-demographic characteristics with EBP utilization among nurses working in ARRH, Ethiopia, 2019 (n = 671).**

| Variables | Categories | EBP Utilization | | COR (95%CI) | AOR (95%CI) |
|---|---|---|---|---|---|
| | | Good N (%) | Poor N (%) | | |
| Sex | Female | 189 (28.2%) | 140 (20.9%) | 1.215 (0.896–1.648) | |
| | Male | 180 (26.8%) | 162 (24.1%) | 1 | |
| Age (year) | 21–30 | 277 (41.3%) | 165 (24.6%) | 2.350 (0.734–7.525) | |
| | 31–40 | 73 (10.9%) | 86 (12.8%) | 1.188 (0.362–3.903) | |
| | 41–50 | 14 (2.1%) | 44 (6.6%) | 0.445 (0.122–1.627) | |
| | 51–60 | 5 (0.7%) | 7(1%) | 1 | |
| Marital status | Single | 154 (23.0%) | 70 (10.4%) | 2.374 (1.693–3.329) | 1.662 (1.089–2.536)* |
| | Married | 215 (32.0%) | 232 (34.6%) | 1 | 1 |
| Level of Education | MSc | 35 (5.2%) | 3 (0.4%) | 16.154 (4.48–58.24) | |
| | BSc | 308 (45.9%) | 263 (39.2%) | 1.622 (0.954–2.757) | |
| | Diploma | 26 (3.9%) | 36 (5.4%) | 1 | |
| Place of graduation | Government | 336 (50.1%) | 262 (39.0%) | 1.554 (0.954–2.533) | |
| | Private | 33 (4.9%) | 40 (6.0%) | 1 | |
| Experience (in years) | <5 | 203 (30.3%) | 122 (18.2%) | 4.453 (2.830–7.009) | 1.849 (1.049–3.257)* |
| | 5–10 | 132 (19.7%) | 89 (13.3%) | 3.970 (2.464–6.396) | 2.227 (1.271–3.900)* |
| | >10 | 34 (5.1%) | 91 (13.6%) | 1 | 1 |
| Working unit | Critical care | 24 (3.6%) | 17 (2.5%) | 1.170 (0.574–2.384) | |
| | Pediatrics | 34 (5.1%) | 40 (6.0%) | 0.704 (0.396–1.251) | |
| | Medical | 100 (14.9%) | 82 (12.2%) | 1.010 (0.642–1.592) | |
| | Surgical | 104 (15.5%) | 78 (11.6%) | 1.105 (0.701–1.742) | |
| | Emergency | 37 (5.5%) | 27(4.0%) | 1.135 (0.619–2.081) | |
| | Ambulatory | 70 (8.6%) | 58 (10.4%) | 1 | |
| Average monthly salary | > 7111 | 55 (8.2%) | 71 (10.6%) | 0.499 (0.313–0.796) | |
| | 5294–7111 | 114 (17.0%) | 85 (12.7%) | 0.864 (0.570–1.310) | |
| | 4650–5294 | 96 (14.3%) | 79 (11.8%) | 0.783(0.510–1.201) | |
| (ETB) | < 4650 | 104 (15.5%) | 67 (10.0%) | 1 | |

**Where: 1** = Reference

** = p< 0.001

* = p< 0.05.

for less than 5 years and 5–10 years were about 85% (AOR = 1.849: 95% CI: 1.049, 3.257) and 2.2 times (AOR = 2.227: 95% CI: 1.271, 3.900) more likely to have good EBP utilization compared to those who had worked more than 10 years respectively (**Table 4**).

The odds of having good EBP utilization were 2 times higher among nurses who had good knowledge about EBP (AOR = 2.044: 95% CI: 1.406, 2.972) than those who had poor knowledge. Similarly, those nurses who had effective communication skills were about 2.5 times (AOR = 2.537: 95% CI: 1.744, 3.689) more likely to have good EBP utilization as compared to their contraries.

Nurses who had taken EBP training as part of their curriculum and or separately were about 3.2 times (AOR = 3.224: 95% CI: 1.957, 5.311) more likely to have good EBP utilization compared with those who did not attend it (**Table 5**).

Besides, nurses who had internet access and availability of evidence-based guideline at the work area were 65.5% and 82.7% more likely to have good EBP utilization as compared to their counterparts (AOR = 1.655: 95% CI: 1.119, 2.448) and (AOR = 1.827: 95% CI: 1.249, 2.673) respectively (**Table 6**).

**Table 5. Binary logistic regression analyses of individual factors to the utilization of evidence-based practice, among nurses working in ARRH, Ethiopia, 2019 (n = 671).**

| Variables | EBP Utilization | | COR (95%CI) | AOR (95%CI) |
|---|---|---|---|---|
| | **Good N (%)** | **Poor N (%)** | | |
| Knowledge about EBP | | | | |
| Good | 253 (37.7%) | 109 (16.2%) | 3.862 (2.800–5.327) | 2.044 (1.406–2.972)** |
| Poor | 116 (17.3%) | 193 (28.8%) | 1 | 1 |
| Attitude towards EBP | | | | |
| Favorable | 210 (31.3%) | 133 (19.8%) | 1.678 (1.235–2.281) | 0.943 (0.637–1.396) |
| Unfavorable | 159 (23.7%) | 169 (25.2%) | 1 | 1 |
| Nurse-patient communication | | | | |
| Effective | 232(34.6%) | 106 (15.8%) | 3.131 (2.281–4.299) | 2.537 (1.744–3.689)** |
| Ineffective | 137 (20.4%) | 196 (29.2%) | 1 | 1 |
| Having basic computer skill | | | | |
| Yes | 275 (41.0%) | 173 (25.8%) | 2.181 (1.573–3.025) | 0.805 (0.517–1.252) |
| No | 94 (14.0%) | 129 (19.2%) | 1 | 1 |
| Having research experience | | | | |
| Yes | 253 (37.7%) | 134 (20.0%) | 2.734 (1.994–3.750) | 1.301 (0.873–1.939) |
| No | 116 (17.3%) | 168 (25.0%) | 1 | 1 |
| The ability of understanding statistical terms | | | | |
| Yes | 246 (36.7%) | 127 (18.9%) | 2.756 (2.012–3.775) | 1.066 (0.697–1.633) |
| No | 123 (18.3%) | 175 (26.1%) | 1 | 1 |
| Ability to interpret research findings | | | | |
| Yes | 230 (34.3%) | 109 (16.2%) | 2.930 (2.137–4.016) | 1.362(0.935–1.985) |
| No | 139 (20.7%) | 193 (28.8%) | 1 | 1 |
| Taking EBP course or training | | | | |
| Yes | 125 (18.6%) | 29 (4.3%) | 4.823 (3.108–7.483 | 3.224 (1.957–5.311)** |
| No | 244 (36.4%) | 273 (40.7%) | 1 | 1 |

**Where:** 1 = Reference

** = p< 0.001

* = p< 0.05.

## Discussion

In this study, 55% (95% CI: 51.2, 58.9) of nurses had good evidence-based practice utilization. This finding is comparable to the study done in Ethiopia [23, 37], Kenya [33], Nigeria [32], and Zambia [30] in which 51.8% and 57.6%, 39.9%, 53.6%, and 54% of nurses utilized research-driven evidence during care provision respectively.

But, this finding is lower when compared to the study conducted in southwest and southern Ethiopia, where 81.1% [38] and 61.5% [39] of nurses utilized EBP during their patient care respectively. This inconsistency could be from a lack of training and knowledge differences. In this study, only 23% of nurses received EBP training. Again, from those studies, it was reported that most of their participants had better knowledge about EBP as a majority, 62.9% [38] and 81.2% [39] of their respondents were familiar with the concept of EBP than this study (34.6%). Hence, knowledgeable nurses are confident and eager to integrate evidence into their practice as compared to their counterparts [37].

This finding is higher than the study done in Ghana, Australia, and Iran whereby only 25.3% [31], one-third [27] and 41% [28] of nurses had utilized evidence-based practice respectively. This could be due to differences in knowledge, attitude, training, and operational

**Table 6. Binary logistic regression analyses of organizational factors to the utilization of evidence-based practice, among nurses working in ARRH, Ethiopia, 2019 (n = 671).**

| Variables | EBP Utilization | | COR (95%CI) | AOR (95%CI) |
|---|---|---|---|---|
| | Good N (%) | Poor N (%) | | |
| Presence of evidence-based guideline in the work area | | | | |
| Present | 274 (40.8%) | 135 (20.1%) | 3.568 (2.577–4.940) | 1.827 (1.249–2.673)* |
| Absent | 95 (14.2%) | 167 (24.9%) | 1 | 1 |
| Internet access at the workplace | | | | |
| Yes | 237 (35.3%) | 117 (17.4%) | 2.839 (2.073–3.888) | 1.655 (1.119–2.448)* |
| No | 132 (19.7%) | 185 (27.6%) | 1 | 1 |
| Computer availability at the work area | | | | |
| Available | 210 (31.3%) | 97 (14.5%) | 2.791 (2.032–3.834) | 1.439 (0.962–2.152) |
| Not available | 159 (23.7%) | 205 (30.6%) | 1 | 1 |
| Library with updated references | | | | |
| Present | 194 (28.9%) | 111 (16.5%) | 1.908 (1.398–2.602) | 0.878 (0.583–1.322) |
| Absent | 175 (26.1%) | 191 (28.5%) | 1 | 1 |
| Presence of a nurse manager who leads and facilitates EBP utilization | | | | |
| Present | 204 (30.4%) | 100 (14.9%) | 2.497 (1.822–3.424) | 1.359 (0.920 2.008) |
| Absent | 165 (24.6%) | 202 (30.1%) | 1 | 1 |
| Presence of enough time at the workplace | | | | |
| Yes | 174 (25.9%) | 88 (13.1%) | 2.170 (1.573–2.993) | 0.851 (0.541–1.339) |
| No | 195 (29.1%) | 214 (31.9%) | 1 | 1 |
| Patient overload (patient to nurse ratio) | | | | |
| $\leq$ 6:1 | 161 (24.0%) | 128 (19.1%) | 1.052 (0.774–1.431) | |
| > 6:1 | 208 (31.0%) | 174 (25.9%) | 1 | |
| Workload | | | | |
| $\leq$39 hours/ week | 157 (23.4%) | 127 (18.9%) | 1.020 (0.750–1.388) | |
| >39 hours/week | 212 (31.6%) | 175 (26.1%) | 1 | |

**Where:** 1 = Reference

** = $p < 0.001$

* = $p < 0.05$.

definition from study to study. Additionally, it could associate that the majority of health workers, including nurses in Africa, did not learn about EBP in nursing schools [21].

After adjusting for possible confounding factors, it was found that marital status, work experience, knowledge, communication skill, internet access, training, and evidence-based guideline availability were significantly associated with EBP utilization.

This study showed that marital status had a significant association with EBP utilization. It was found that nurses who were single in marital status were 66.2% more likely (AOR = 1.662, 95% CI: 1.089, 2.536) to have good EBP utilization than married nurses. This is in contrast with the study of Kenya [49] in which marital status has no relationship with the utilization of EBP but analogous with another study [39]. The reason could be because married nurses have added duties at home, which puts them in an extra workload. Due to this, married nurses might not search, appraise, and integrate evidence properly.

In this study, work experience was significantly associated with the utilization of EBP. Nurses who had worked for less than 5 years and 5–10 years were about 1.85 (AOR = 1.849, 95% CI: 1.049, 3.257) and 2.2 times (AOR = 2.227, 95% CI: 1.271, 3.900) more likely to have good EBP utilization compared to those who had worked more than 10 years respectively.

This finding is consistent with the study from Iran [50] but, reverse to the study from Ethiopia [23] and Norway [24] that stated work experience had no association and having less work experience had a preventive association to EBP respectively. This discrepancy may be due to differences in the study period. But, the possible reason for the above finding could be that those nurses who have ten or more years of working experience may not get the opportunity of learning EBP at nursing school since EBP is a new approach in the region and may not update their knowledge as science is dynamic. Additionally, younger nurses are better at valuing EBP and elders were tending to use self-experience [24].

This study revealed that having good knowledge about EBP was 2 times (AOR = 2.044, 95% CI: 1.406–2.972) more likely to have good EBP utilization compared to having poor knowledge about EBP. The possible explanation for this could be knowledge about EBP may increase their appraisal skills and give more confidence in utilizing EBP. Additionally, a large proportion of our respondents were in the younger age group (21–30 years), who might be active in information sharing. This finding is supported by results from Ethiopia [23, 37, 38], Malaysia [51], and the US [52] whereby nurses who knew about evidence-based practice concepts were more likely to utilize evidence-based practice than those who did not know.

This study was added a new predictor variable, communication skill, and has got a significant association with EBP utilization. Those nurses who had effective communication with the patient were 2.5 times (AOR = 2.537, 95% CI: 1.744–3.689) more likely to have good EBP utilization as compared to those who had ineffective communication. The reason for this finding might be since having effective communication with the patient can prevent the missing and wrong interpretation of the patient's important information and creates a trustworthy environment between the two. This may have contributed to integrating the patient's values and preferences. Additionally, this might be due to the notion that nurses' communication skill is the key to the quality of patient care [48].

Similar to studies conducted in Ethiopia [25] and Iran [28], this study also found out that training about EBP was significantly associated with the utilization of EBP. It was observed that nurses who attend EBP training either as part of their curriculum or separately are 3.2 times (AOR = 3.224, 95% CI: 1.957–5.311) more likely to have good EBP utilization compared to those who do not attend it. The reason could be that training may help nurses to be clear more about the steps of EBP. Again, a study found that those respondents who had attended EBP training got themselves more comfortable in integrating EBP into their practice [53].

Internet access was also found one of the factors significantly associated with EBP utilization. Nurses who had internet access at the work area were 65.5% more (AOR = 1.655, 95% CI: 1.119–2.448) likely to have good EBP utilization as compared to those who hadn't. This finding is in agreement with studies done in Ethiopia [25], Ghana [31], Uganda [54], Norway [24], Malaysia [51], and the US [52]. This is because EBP utilization needs some prerequisites like searching over the internet and easy access to online EBP resources may simplify the use of evidence-based practice [1].

Furthermore, this study found that evidence-based guideline availability in the work area had a significant association with evidence-based practice utilization. Nurses who had guidelines at the work were 1.8 times more likely to have good EBP utilization as compared to their counterparts (AOR = 1.827, 95% CI: 1.249–2.673). This finding is consistent with studies done in Ethiopia [37] and USA [26] that state lack of updated guidelines and organizational factors like the absence of resources to be accessed in the work area is the hindering factor from utilizing EBP. The reason may be that if evidence-based guidelines are available in the work area, nurses could refer them at the bedside and provide scientific-based care without delay.

Our study included all referral hospitals in the region. These findings therefore could be generalizable to nurses working in similar settings and incorporating one important predictor

variable, communication skill makes this study stronger. But this study might encounter response bias as using a self-administered questionnaire was considered as a limitation of this study and as per the researchers' literature review, this study was done on certain variables. It would have been worthy if role at the hospital and shifting type had been determined.

## Conclusions and recommendations

### Conclusion

Even though more than half (55%) of nurses had good utilization, evidence-based practice in this study was found to be low as compared to studies conducted in other parts of Ethiopia. Being single in marital status, having less work experience, good EBP knowledge, effective communication skills, taking EBP training, internet access, and having evidence-based guidelines at the work area were predictor variables to EBP utilization.

### Recommendations

Based on the finding of this study, the following recommendations are made:

**To Federal Ministry of Health (FMOH):**

i. The FMOH should try to build nurses' EBP competencies either by giving short-term EBP training or incorporating EBP as part of the curriculum.

**To Hospital administrator:**

i. It is better to avail evidence-based guidelines in the work area and improve the facility's internet access.

ii. It is good to orient on how to utilize EBP for early graduated nurses.

**To Further researchers:**

i. Although this study had tried to assess the utilization of EBP, the researcher believes that the triangulation of methods (e.g., observation) could help in determining real utilization.

**To Nurses:**

i. Communication skill was emerging as a predicting variable to EBP. So, nurses need to give attention to it along with their EBP.

## Supporting information

**S1 Checklist. STROBE checklist.**
(DOC)

**S1 File. Data set on evidence-based practice utilization.**
(SAV)

## Acknowledgments

First, we would like to forward our heartfelt gratitude to all Referral Hospital Matrons in the region for their valuable study setting information. Secondly, we want to thank all supervisors, data collectors, and study participants. Finally, we would like to give our greatest thanks to University of Gondar, School of Nursing for offering the opportunity to carry out this study.

## Author Contributions

**Conceptualization:** Zewdu Bishaw Aynalem, Kassahun Gebeyehu Yazew, Mignote Hailu Gebrie.

**Data curation:** Zewdu Bishaw Aynalem.

**Formal analysis:** Zewdu Bishaw Aynalem, Kassahun Gebeyehu Yazew, Mignote Hailu Gebrie.

**Funding acquisition:** Zewdu Bishaw Aynalem.

**Investigation:** Zewdu Bishaw Aynalem.

**Methodology:** Zewdu Bishaw Aynalem, Kassahun Gebeyehu Yazew, Mignote Hailu Gebrie.

**Validation:** Kassahun Gebeyehu Yazew, Mignote Hailu Gebrie.

**Visualization:** Zewdu Bishaw Aynalem, Kassahun Gebeyehu Yazew, Mignote Hailu Gebrie.

**Writing – original draft:** Zewdu Bishaw Aynalem.

**Writing – review & editing:** Zewdu Bishaw Aynalem, Kassahun Gebeyehu Yazew, Mignote Hailu Gebrie.

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
