## [Editor Report · Decision Letter 0]

15 Apr 2020

PONE-D-20-08386

Evidence-Based Practice Utilization and Associated Factors among Nurses working in Amhara Region Referral Hospitals, Ethiopia.

PLOS ONE

Dear Mr. Aynalem,

Thank you for submitting your manuscript to PLOS ONE. After careful consideration, we feel that it has merit but does not fully meet PLOS ONE’s publication criteria as it currently stands. Therefore, we invite you to submit a revised version of the manuscript that addresses the points raised during the review process.

We would appreciate receiving your revised manuscript by May 30 2020 11:59PM. To enhance the reproducibility of your results, we recommend that if applicable you deposit your laboratory protocols in protocols.io, where a protocol can be assigned its own identifier (DOI) such that it can be cited independently in the future. For instructions see: http://journals.plos.org/plosone/s/submission-guidelines#loc-laboratory-protocols

We look forward to receiving your revised manuscript.

Kind regards,

Tim Schultz

Academic Editor

PLOS ONE

Journal Requirements:

1. We note that you have included the questionnaires used in the study as Supporting Information. Please ensure that you have permissions to publish these scales with a CC BY license. We would suggest that you remove any questions that you do not have permission to publish and include a reference instead. For more detail please see here: https://journals.plos.org/plosone/s/licenses-and-copyright.

2. Please upload a copy of Figure 1, to which you refer in your text on page 13. If the figure is no longer to be included as part of the submission please remove all reference to it within the text.

3. We note you have included a table to which you do not refer in the text of your manuscript. Please ensure that you refer to Table 1 in your text; if accepted, production will need this reference to link the reader to the Table.

Additional Editor Comments (if provided):

Thank-you for your submission “Evidence-Based Practice Utilization and Associated Factors among Nurses working in Amhara Region Referral Hospitals, Ethiopia”.

Based on an initial review of the manuscript, it is not currently suitable for peer review. There are a number of issues to be addressed that are documented on the attached PDF. Please note that while it is essential to address these issues for the manuscript to be subsequently sent for peer review, resubmitting the manuscript will not automatically guarantee peer review.

As annotated in the PDF, the main issues are:

• Inappropriate referencing from blog posts

• Poor English expression. A number of incidences are marked up, but the whole manuscript requires editing for English expression.

• Lack of clarity about the tools that were used to measure the outcome and the independent variables. How were they derived? What are the psychometric properties? Have they been validated in other prior studies? What is the justification for the use of cut-off between what is good and bad EBP?
---

## [Author Response · Author response to Decision Letter 0]

23 Jul 2020

Reviewer # 1: We have incorporated all of your suggestions in to our revision. All are very constructive.

---

## [Decision Letter · Decision Letter 1]

8 Oct 2020

PONE-D-20-08386R1

Evidence-Based Practice Utilization and Associated Factors among Nurses working in Amhara Region Referral Hospitals, Ethiopia.

PLOS ONE

Dear Dr. Aynalem,

Thank you for submitting your manuscript to PLOS ONE. After careful consideration, we feel that it has merit but does not fully meet PLOS ONE’s publication criteria as it currently stands. Therefore, we invite you to submit a revised version of the manuscript that addresses the points raised during the review process.

We look forward to receiving your revised manuscript.

Kind regards,

Tim Schultz

Academic Editor

PLOS ONE

Additional Editor Comments (if provided):

Please see comments from two reviewers below. All comments need to be addressed. Additionally please ensure that the manuscript complies with the STROBE checklist for cross sectional studies and upload a completed STROBE checklist.

Reviewers' comments:

Reviewer's Responses to Questions

**Comments to the Author**

1. If the authors have adequately addressed your comments raised in a previous round of review and you feel that this manuscript is now acceptable for publication, you may indicate that here to bypass the “Comments to the Author” section, enter your conflict of interest statement in the “Confidential to Editor” section, and submit your "Accept" recommendation.

Reviewer #1: (No Response)

Reviewer #2: (No Response)

2. Is the manuscript technically sound, and do the data support the conclusions?

Reviewer #1: Yes

Reviewer #2: Partly

3. Has the statistical analysis been performed appropriately and rigorously? 

Reviewer #1: Yes

Reviewer #2: No

4. Have the authors made all data underlying the findings in their manuscript fully available?

Reviewer #1: Yes

Reviewer #2: Yes

5. Is the manuscript presented in an intelligible fashion and written in standard English?

Reviewer #1: Yes

Reviewer #2: No

6. Review Comments to the Author

Reviewer #1: (No Response)

Reviewer #2: Thank-you for the opportunity to review R1 of this manuscript. While it does have its strengths, including a high response rate and a large sample size, there are some issues that need to be addressed. My main criticism reflects my comment from the first version of the manuscript in relation to the use of cut-off scores to dichotomise a variable that could be considered continuous (eg EBP utilisation on a range of 6-30) into eg low and high, or poor and good. This analysis has two implications: with a cut-off of the median 16, a score of 15 is considered equivalent to a score of 10 and 5. A lot of variation in the data is consequently lost. Second, if the cut-off is 16 and above, then of course a little more than 50% of the responses are going to fall in this category. So, the main finding that 55% of nurses had good EBP utilisation is completely meaningless.

I would recommend using multiple regression on the total EBP utilisation score (6-30) instead. And this suggestion also applies to the predictors Knowledge, Attitude and Communication Skill. They could all be added as continuous predictors in a multiple regression.

There are a range of other criticisms of the paper included on the PDF of the manuscript. I believe that all should be addressed including:

• Structure of the tables (there are too many tables)

• Numerous instances of poor English expression

• Remove the general objectives

• Clarifying why some of the variables don’t have adjusted odds ratios given (eg see comment on page 15)

7. PLOS authors have the option to publish the peer review history of their article (what does this mean?). If published, this will include your full peer review and any attached files.

Reviewer #1: No

Reviewer #2: **Yes: **Tim Schultz

---

## [Author Response · Author response to Decision Letter 1]

22 Nov 2020

Reviewer 1: We have incorporated all of your suggestions into the revision. All the comments are very helpful.

Reviewer 2: We have incorporated all of your suggestions into the revision. All the comments are very helpful.

---

## [Decision Letter · Decision Letter 2]

4 Feb 2021

PONE-D-20-08386R2

Evidence-Based Practice Utilization and Associated Factors among Nurses working in Amhara Region Referral Hospitals, Ethiopia.

PLOS ONE

Dear Dr. Zewdu Bishaw Aynalem,

Thank you for submitting your manuscript to PLOS ONE. After careful consideration, we feel that it has merit but does not fully meet PLOS ONE’s publication criteria as it currently stands. Therefore, we invite you to submit a revised version of the manuscript that addresses the points raised during the review process.

We look forward to receiving your revised manuscript.

Kind regards,

Sharon Mary Brownie

Academic Editor

PLOS ONE

 Editor Comments 

The manuscript is much improved, however, the need for some minor revisions remain. Please address the identified issues.

Reviewers' comments:

Reviewer's Responses to Questions

**Comments to the Author**

1. If the authors have adequately addressed your comments raised in a previous round of review and you feel that this manuscript is now acceptable for publication, you may indicate that here to bypass the “Comments to the Author” section, enter your conflict of interest statement in the “Confidential to Editor” section, and submit your "Accept" recommendation.

Reviewer #1: All comments have been addressed

Reviewer #3: All comments have been addressed

2. Is the manuscript technically sound, and do the data support the conclusions?

Reviewer #1: Yes

Reviewer #3: Yes

3. Has the statistical analysis been performed appropriately and rigorously? 

Reviewer #1: Yes

Reviewer #3: Yes

4. Have the authors made all data underlying the findings in their manuscript fully available?

Reviewer #1: Yes

Reviewer #3: Yes

5. Is the manuscript presented in an intelligible fashion and written in standard English?

Reviewer #1: Yes

Reviewer #3: Yes

6. Review Comments to the Author

Reviewer #1: (No Response)

Reviewer #3: I will like to thank the researchers for addressing the comments of the reviewers. However, there remains some discretionary but important comments to address that will strengthen the manuscript.

Please see comments below:

1) On page 6, under the heading Study design, setting, and participants, the authors need to rephrase the sentence "During the study period, there were a total of 458, 390, 254, 140 and 223". The authors should provide the total number of nurses/participants in the facility and then provide the breakdown of the participants per unit (for example, 458 participants in the General ward, 390 in the pediatric ward etc.)

2) On page 8, paragraph 2, the sentence "The principal investigator was made the overall supervision..." needs to be revised.

3) On page 21, the recommendation for the Hospital administrator, bullet point i), please revise the sentence "It is better to avail of evidence-based guidelines". I believe the word 'of' should be removed.

Otherwise, the manuscript has been significantly improved.

7. PLOS authors have the option to publish the peer review history of their article (what does this mean?). If published, this will include your full peer review and any attached files.

Reviewer #1: No

Reviewer #3: No

---

## [Author Response · Author response to Decision Letter 2]

28 Feb 2021

Reviewer # 1: Thank you for your genuine comments.

Reviewer #3: Thank you. We hope all the comments are corrected.

---

## [Editor Report · Decision Letter 3]

8 Mar 2021

Evidence-Based Practice Utilization and Associated Factors among Nurses working in Amhara Region Referral Hospitals, Ethiopia.

PONE-D-20-08386R3

Dear Dr.Zewdu Bishaw Aynalem,

We’re pleased to inform you that your manuscript has been judged scientifically suitable for publication and will be formally accepted for publication once it meets all outstanding technical requirements.

Kind regards,

Sharon Mary Brownie

Academic Editor

PLOS ONE

---

## [Editor Report · Acceptance letter]

12 Mar 2021

PONE-D-20-08386R3 

Evidence-Based Practice Utilization and Associated Factors among Nurses working in Amhara Region Referral Hospitals, Ethiopia. 

Dear Dr. Aynalem:

I'm pleased to inform you that your manuscript has been deemed suitable for publication in PLOS ONE. Congratulations! Your manuscript is now with our production department. 

Kind regards, 

on behalf of

Professor Sharon Mary Brownie 

Academic Editor

PLOS ONE